energy/environmental engineering/chemical engineering

coal pile, self-heating characteristics, Frank–Kamenetskii theory, porosity, volume, coal particle size

**Authors for correspondence:**
Yongjun Wang
e-mail: wyj917@126.com
Xiaoming Zhang
e-mail: xmzhang7@126.com

# Effects of temperature gradient and particle size on self-ignition temperature of low-rank coal excavated from inner Mongolia, China

Yongjun Wang[1], Xiaoming Zhang[1,2], Hemeng Zhang[1] and Kyuro Sasaki[3]

[1]College of Mining Engineering, Liaoning Technical University, Fuxin 123000, People's Republic of China
[2]Institute of Engineering and Environment, Liaoning Technical University, Huludao 125000, People's Republic of China
[3]Faculty of Engineering, Kyushu University, 744 Motooka, Nishiku, Fukuoka 819-0385, Japan

YW, 0000-0003-3861-2400

This study investigates the effects of temperature gradient and coal particle size on the critical self-ignition temperature $T_{CSIT}$ of a coal pile packed with low-rank coal using the wire-mesh basket test to estimate $T_{CSIT}$ based on the Frank–Kamenetskii equation. The values of $T_{CSIT}$, the temperature gradient and the apparent activation energy of different coal pile volumes packed with coal particles of different sizes are measured. The supercriticality or subcriticality of the coal is assessed using a non-dimensional index $I_{HR}$ based on the temperature gradient at the temperature cross-point between coal and ambient temperatures for coal piles with various volumes and particle sizes. The critical value $I_{HRC}$ at the boundary between supercriticality and subcriticality is determined as a function of pile volume. The coal status of supercritical or subcritical can be separated by critical value of $I_{HR}$ as a function of pile volume. Quantitative effects of coal particle size on $T_{CSIT}$ of coal piles are measured for constant pile volume. It can be concluded that a pile packed with smaller coal particles is more likely to undergo spontaneous combustion, while the chemical activation energy is not sensitive to coal particle size. Finally, the effect of coal particle size on $T_{CSIT}$ is represented by the inclusion of an extra term in the equation giving $T_{CSIT}$ for a coal pile.

# 1. Introduction

Coal has long been the primary fossil fuel for industrial applications worldwide, because of its huge reserves and mining convenience. Whatever the mining method, coal is transported to the ground in large quantities, and because of the intrinsic properties of coal, this presents hazards in all processing stages after mining [1]. These hazards include spontaneous combustion resulting from the complex physical and chemical structure of coal and its propensity to oxidization [2–5]. The spontaneous combustion of coal starts from slow physico-chemical reaction processes in the low-temperature range, culminating in a final stage when red-hot spots can be found on the surfaces of coal piles or pillars.

Numerous studies have demonstrated that spontaneous combustion of coal involves a number of stages with different kinetic characteristics [6,7]. During the past few decades, numerous experimental and numerical studies have been performed with the aim of preventing spontaneous combustion in industrial processes by interfering with the process in a variety of ways at different stages [8,9]. Some of these approaches have focused on the study of the combustion products of the low-temperature oxidation process. From an examination of the mechanism of oxidation of coal of different particle sizes in the temperature range below 100°C, Wang *et al.* [10] proposed a temperature threshold for thermal decomposition in low-temperature oxidation. Using techniques of organic chemistry and quantum chemistry, Xu *et al.* [11] studied heat generation by oxygen-containing functional groups during spontaneous combustion. They found that the presence of such groups is critical to heat production during the early stages of combustion and that this explained why the coal temperature rises slowly in this stage. Observations of raw and pyrolysed coal samples have shown that the active sites are influenced by the content of oxygen-containing functional groups, pyrolysis temperature, oxidation temperature, particle size and pyrolysis–oxidation times. Li *et al.* [12] proposed that two reaction processes are involved in coal oxidation: oxidation of active sites after coal pyrolysis and subsequent thermal decomposition with the generation of functional groups.

We believe that laboratory experiments are also important for analysis of the self-heating of coal, prediction of spontaneous combustion and design of fire extinguishing methods [13,14]. Beamish & Hamilton [15] studied coal sample oxidation under adiabatic low-temperature conditions. They defined the relevant characteristics of coal samples using the $R_{70}$ value, which is an indicator of the reactivity of coal to oxygen that induces self-heating. Subsequently, they also assessed the effects of moisture and ash on the self-heating rate of coal samples. Xu *et al.* [16] established a platform for the study of spontaneous combustion of coal with the aim of determining the temperature changes in a coal pile, and deduced the degree of oxidation of loosely packed coal piles under different low-temperature conditions. Subsequently, they quantitatively analysed the spontaneous combustion characteristics of different coals. Using laboratory measurements with small volumes of coal particles, Kim *et al.* [17] proposed a low-temperature reaction index based on the variations in temperature and temperature gradient in an oxygen-rich atmosphere. Peng *et al.* [18] presented a short-term prediction scheme to prevent spontaneous combustion of coal piles by establishing a grey model to analyse the sensitivities of parameters such as moisture content, ash content, shape and temperature distribution.

As described above, previous experimental and numerical simulation studies have focused mainly on analysing the oxidation characteristics under adiabatic thermal conditions without consideration of pile volume. Therefore, even if different sizes of coal particles are used in the experiments, the results do not give an adequate representation of the effects that appear under thermal adiabatic conditions. For coal pile management, the effect of coal particle size has to take account of the relation between the critical self-ignition temperature $T_{CSIT}$ and pile volume.

Our previous work [19] considered primarily the effect of coal pile volume on $T_{CSIT}$ while taking account of natural convection flow in a coal pile. We presented the following equation for $T_{CSIT}$ (°C) as a logarithmic function of the coal basket volume $V$ (m$^3$):

$$T_{CSIT} = -12.73 \cdot \log(V) + 81.5. \tag{1.1}$$

This equation shows that $T_{CSIT}$ becomes lower with increasing $\log(V)$. For example, $T_{CSIT}$ is reduced by about 25°C when the volume of a pile is enlarged by two orders of magnitude. We also confirmed that the coal pile volume, porosity, and thermal diffusivity influenced $T_{CSIT}$ by their effects on the thermal equilibrium state. Furthermore, we explained that the natural convection flow through a coal pile is a key factor maintaining heat balance, because it provides oxygen to the pile for heat generation and takes heat to the outside, especially when the coal pile central temperature becomes equal to the ambient temperature (hereinafter, this is called the temperature cross-point, TCP). This is because the balance between heat generation and dissipation is very sensitive to natural convection, and it is also affected by

coal particle size through permeability, which is proportional to the square of the size according to the Kozeny–Carman equation [20]. Therefore, coal particle size is one of the major parameters affecting coal oxidation, heat transfer and drying of moisture through natural convection flow in a coal pile. However, the quantitative effect of particle size on $T_{\mathrm{CSIT}}$ has not been established systematically for different coal pile volumes. Furthermore, it is expected that self-ignition of coal may also be related to the coal temperature gradient at the TCP.

In this study, the effects of coal particle size in coal basket piles and the temperature gradient at the TCP on the characteristics of coal self-ignition are investigated experimentally using the wire-mesh basket (WMB) test with a low-rank coal sample excavated from Inner Mongolia, China, with particular consideration of changes in $T_{\mathrm{CSIT}}$.

# 2. Heat balance in coal piles

## 2.1. Frank–Kamenetskii equation

The Arrhenius equation describes the oxidative exothermic process with temperature change [21]. During the self-heating process of the coal, the kinetic temperature change under thermal adiabatic conditions can be expressed as follows:

$$Q = \rho \cdot C_p \cdot \frac{dT_C}{dt} = A \cdot \exp\left[-\frac{E}{RT_K}\right], \tag{2.1}$$

where $Q$ is the volumetric heat generation rate (W m$^{-3}$), $T_C$ (°C) is the coal temperature, $t$ is the elapsed time (s), $\rho$ is the coal density (kg m$^{-3}$), $C_p$ is the specific heat capacity of the coal (J kg$^{-1}$ °C$^{-1}$), $T_K$ (= $T_C$ + 273) is the absolute temperature of the coal (K), $A$ is a frequency factor (W m$^{-3}$), $E$ is the apparent activation energy (J mol$^{-1}$) and $R$ is the ideal gas constant (8.314 J K$^{-1}$ mol$^{-1}$).

The apparent activation energy $E$ is an important parameter indicating the reactivity of coal oxidation. Suppose that the heat generation rate $Q$ is expressed by the Arrhenius equation; then equation (2.1) can be transformed as follows:

$$\ln\left(\frac{dT_C}{dt}\right) = \ln\left(\frac{Q \cdot A}{\rho \cdot C_p}\right) - \frac{E}{R} \cdot \frac{1}{T_K}. \tag{2.2}$$

By plotting $\ln(dT_C/dt)$ against $-T_K^{-1}$, $\ln(QA/\rho/C_p)$ and $E/R$ are obtained from the values of the intercept and slope of the linear equation, respectively. The coal density $\rho$ and specific heat capacity of coal $C_p$ were measured before the experiment. The coal temperature $T_K$ can be measured during the experiment. Finally, the heat generation rate $Q$ and the frequency factor $A$ can be obtained from equations (2.1) and (2.2) [19].

## 2.2. Definition and estimation of $T_{\mathrm{CSIT}}$

Figure 1 shows a schematic of a cubic coal pile with length $L_S$ (m), vertical distance $z$ from pile bottom (m), internal coal temperature $T_C$ (°C) and environmental (outer) temperature $T_E$ (°C). The critical self-ignition temperature $T_{\mathrm{CSIT}}$, which is the value of the ambient (environmental) temperature $T_E$ at the boundary between supercritical (ignition) and subcritical (stabilized to the environment) conditions, is estimated from the coal temperature–time, $T_C$ (°C)–$t$ (s), curves at the pile centre, as shown in figure 2a. It can be related to the pile volume $V$ on the basis of the Frank–Kamenetskii (F–K) model [19]. Meanwhile, as shown in figure 2b, the heating rate of the coal, i.e. the time gradient of the temperature, $dT_C/dt$ (°C s$^{-1}$), can be obtained from the difference between the supercritical and subcritical temperature profiles. Hence, $T_{\mathrm{CSIT}}$ is estimated by interpolating between the two curves.

# 3. Experimental apparatus and coal samples

## 3.1. Experimental apparatus

Figure 3 presents photographs of the experimental apparatus used in this study. The coal samples were heated in cubic WMBs placed in a constant-temperature air chamber, controlled within an error of 1°C. The coal temperatures at the centre of the WMB were recorded and analysed by plotting the data based on equation (2.1). For the same coal samples, differential thermal analysis (DTA) was performed using a

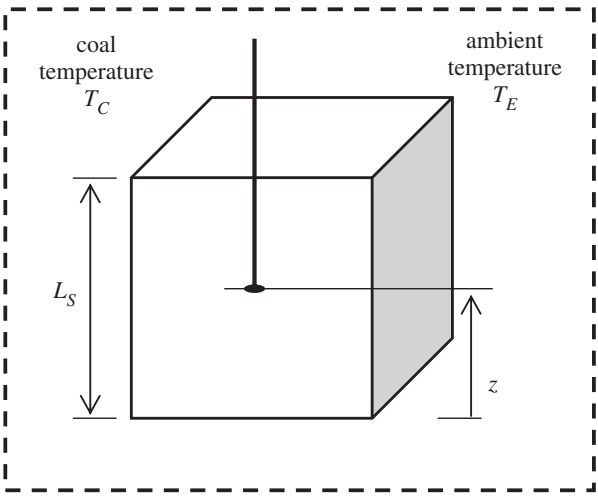

**Figure 1.** Cubic coal pile.

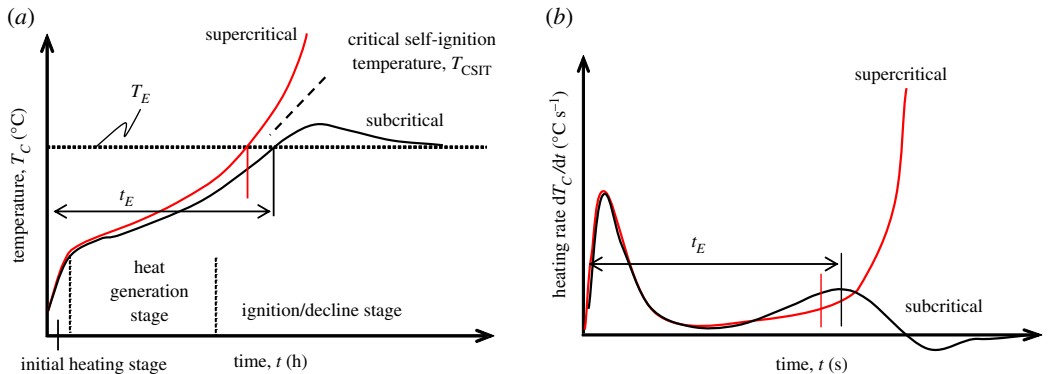

**Figure 2.** Estimation of $T_{CSIT}$ based on the pile temperature profile at constant ambient temperature: (*a*) temperature profile; (*b*) heating rate.

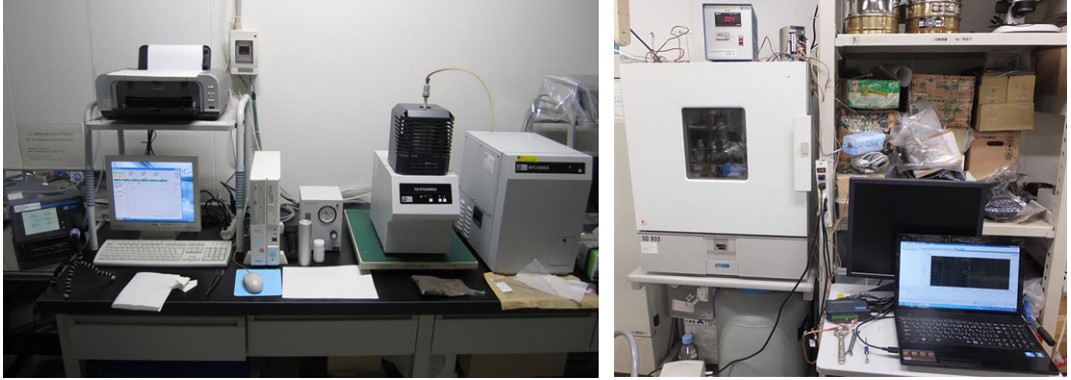

**Figure 3.** DTA system and constant-temperature chamber.

thermogravimetry (TG)–DTA system (Perkin Elmer TAC 7/DX) with a 10°C min$^{-1}$ heating rate and 50 cm$^{3}$ min$^{-1}$ air flow rate for a 30 mg sample.

Figure 4 shows the measurement system of the WMB test for investigating the $T_{CSIT}$ of coal samples [19]. K-type thermocouples were employed to probe the temperature changes in different positions of the cubic WMB, with the coal pile height from the basket bottom ($z = 0$) being denoted by $L_S$ (cm). Three sizes of cubic WMB (5.0, 10.0 and 15.0 cm) were used to perform the tests.

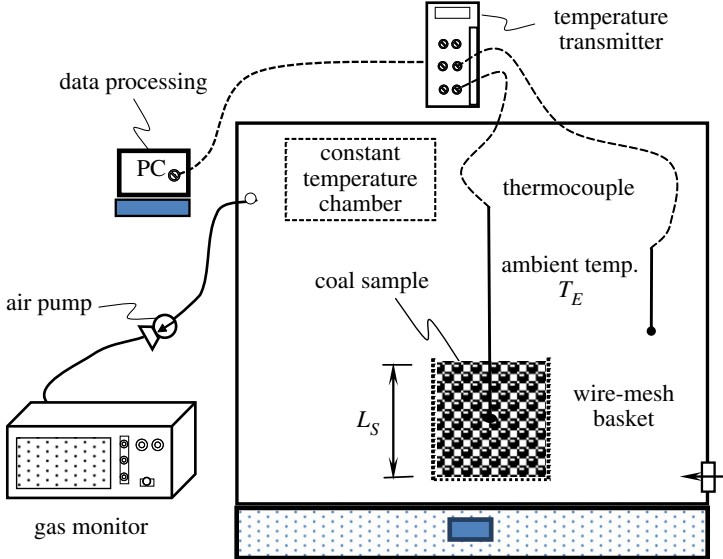

**Figure 4.** Experimental apparatus and temperature and gas monitoring system for the WMB test.

**Table 1.** Properties of coal sample.

| analysis | content | original wet | dry |
|---|---|---|---|
| proximate analysis | ash (%) | 8.3 | 33.42 |
| | fixed carbon (%) | 24.7 | 34.92 |
| | volatile matter (%) | 39.8 | 31.66 |
| | moisture (%) | 27.2 | 0 |
| ultimate analysis | C (%) | 70.44 | 32.42 |
| | H (%) | 4.91 | 2.26 |
| | O (%) | 21.39 | 9.76 |
| | N (%) | 1.35 | 0.62 |
| | S (%) | 1.91 | 0.53 |
| physical properties | density $\rho$ (kg m$^{-3}$) | 1300 | 1560 |
| | heat capacity $C_p$ (J kg$^{-1}$ °C$^{-1}$) | 2720 | 1740 |
| | heat conductivity $\lambda$ (W m$^{-1}$ °C$^{-1}$) | 0.21 | 0.07 |
| | thermal diffusivity $\alpha$ (m$^2$ s$^{-1}$) | $5.94 \times 10^{-8}$ | $2.58 \times 10^{-8}$ |

## 3.2. Properties of coal sample

The low-rank coal used in this study was excavated from Inner Mongolia. Proximate and ultimate analyses of the sample indicate that the coal has high volatile content (39.8%) and moisture content (27.2%), typical of low-rank coal (table 1). The coal sample was prepared by crushing fresh low-rank coal and sieving it into six particle sizes (0.5, 1.0, 3.0, 5.0, 7.5 and 10 mm) to investigate the effects of particle size on $T_{\text{CSIT}}$ and the activation energy $E$.

## 3.3. Differential thermal analysis curve and heating value

The coal sample DTA curve is shown in figure 5. Under a constant heating rate and airflow, the first endothermic peak (at which moisture evaporates and absorbs heat) appears at 124°C. The temperatures at which the heat flow $Q = 0$ are 166°C and 657°C. These results demonstrate that the heat release from and the heat absorption by the coal sample are balanced. Therefore, the heating

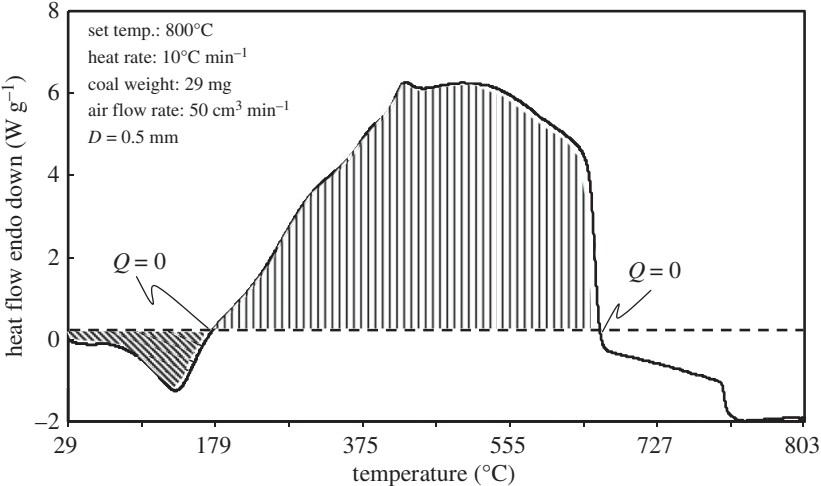

**Figure 5.** DTA curve of the coal sample used in this study (original wet).

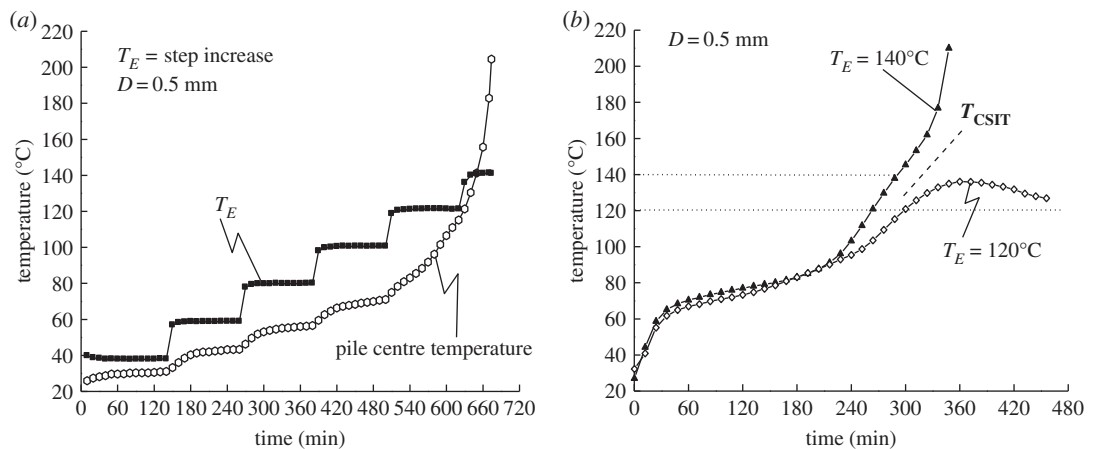

**Figure 6.** Examples of temperature profile at pile centre versus time ($L_S = 5$ cm, initial coal temperature = 25°C): (*a*) stepwise increase in ambient temperature (20°C every 2 h); (*b*) instantaneous setting of a constant ambient temperature from room temperature.

value of the coal sample can be estimated as 14 400 kJ kg$^{-1}$ (= 3440 kcal kg$^{-1}$) by integrating the heat generation between the two points where $Q = 0$ in the dry base condition.

However, according to the TG–DTA results, the coal is in an endothermic condition below 166°C. This temperature is much higher than the $T_{\mathrm{CSIT}}$ values that we have reported previously [19]. In conclusion, it should be emphasized that the TG–DTA result is a thermal characteristic of the coal and does not fully demonstrate self-ignition properties, because it is affected by the heating rate and does not include the effects of coal pile volume.

The endothermic phenomenon, as shown in figure 5, does not have a large effect on the supercriticality or subcriticality of a coal pile based on the heat balance in the pile, because the endothermic heat is much less than the exothermic heat and is generated in the lower-temperature range owing to evaporation of coal moisture.

# 4. Measurement results and discussion

## 4.1. Temperature profile at the centre of a cubic pile for different ambient temperatures

The temperature changes at the pile centre in the 5 cm basket were measured for different profile settings for the ambient air temperature. In the first profile, the ambient air temperature increased from room temperature (25°C) by 20°C every 2 h, as shown in figure 6*a*. In this case, heat conduction from the

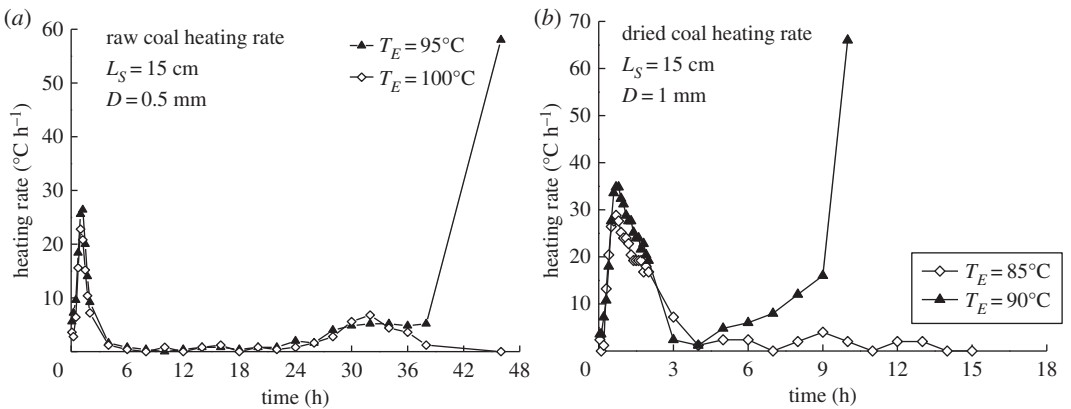

**Figure 7.** Comparison of heating rate–time profiles between (a) original wet and (b) dried coal samples.

ambient air contributes to the increase in coal temperature in the initial heating stage before 60°C, because the coal temperature rises with rising ambient temperature. From 60°C upwards, the rate of increase of coal temperature becomes slower than that of the ambient temperature, because heat conducted from the ambient air is used also for coal drying, which needs latent heat for moisture evaporation. After a temperature of 80°C has been reached, moisture evaporation from the coal sample is almost complete, and oxidation of the coal starts with further increase in coal temperature. After the coal temperature has reached 120°C, oxidation chain reactions induce a state of self-heating, with an exponential increase in heating rate.

As shown in figure 6a, the $T_{CSIT}$ range of the coal sample is between 120°C and 140°C for the present test condition. However, this temperature profile of the ambient temperature using a stepwise increase in ambient air temperature is recognized as not the best methodology to investigate the self-ignition characteristics of coal, because it requires over 10 h, and the temperature step becomes a parameter that is similar to the heating rate in the TG–DTA test.

On the other hand, figure 6b shows the results using a method in which the coal pile is instantaneously set at a constant ambient temperature $T_E$. For the case $T_E = 120$°C, the centre of the coal pile reaches a maximum temperature of 140°C; however, it falls to the ambient temperature (120°C) after $t = 360$ min. Nevertheless, for the case $T_E = 140$°C, the coal temperature rise is accelerated after crossing with $T_E = 140$°C at $t = 285$ min. These results also clearly demonstrate that $T_{CSIT}$ of the coal sample is between 120°C and 140°C. It is easy to understand these results, and to estimate $T_{CSIT}$ [19], because the $T_{CSIT}$ is based on just one temperature parameter (the ambient temperature $T_E$) and the coal pile volume. In this study, the $T_{CSIT}$ is determined based on instantaneous setting of the ambient temperature, as shown in figure 6b.

## 4.2. Effects of moisture on coal heating rate

The effects of moisture on the self-heating characteristics of a coal pile were also investigated. The characteristics of the original and dried coal samples are listed in table 1. To determine the effect of moisture on the coal pile heating rate under given conditions, temperature–time curves of the original and dried coal samples were measured for the same volume in a uniform heating environment. Figure 7a,b shows the heating rate–time profiles of the original and dried coal samples, respectively, in the 15 cm WMB. As can be seen, when the temperature is stable, the time for the dried coal sample ($t = 4$–4.5 h) is much shorter than that for the original wet coal ($t = 6$–24 h), and the dried coal starts to self-ignite earlier. The moisture in the coal is indirectly reflected in the change in the overall heat budget during the self-heating process. For the dried coal sample, $T_{CSIT}$ decreases by approximately 7° C, as shown in figure 7. In the same temperature environment, the inhibitory effect of moisture evaporation on temperature rise is significantly weaker after drying. In essence, drying enhances the tendency of the coal to spontaneously combust, owing to the decrease in the $T_{CSIT}$ of the coal pile. During the preheating process, the greater temperature difference between the coal pile and the environment induces a more intense and more rapid heating effect. However, after the drying period of the raw coal sample is complete, the self-ignition process is almost the same as that of the dried coal (figure 7).

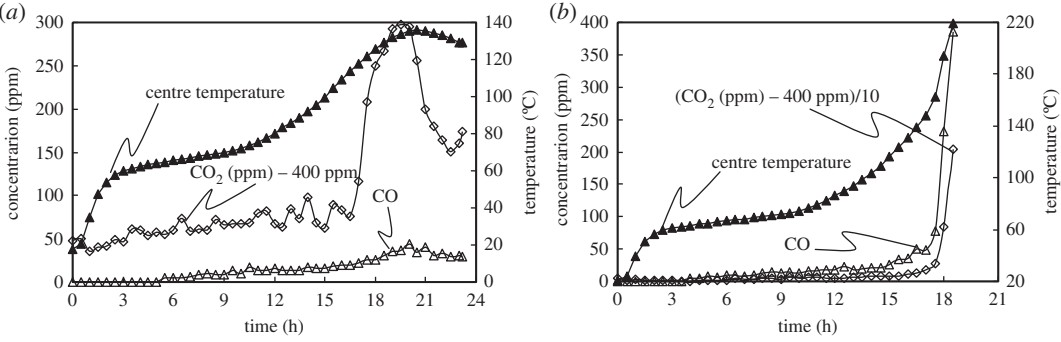

**Figure 8.** Gas concentration and heat generation rate for different ambient temperatures ($L_S = 10$ cm, $D = 5$ mm, original coal): (a) $T_E = 115°C$ (subcritical); (b) $T_E = 120°C$ (supercritical).

When the coal temperature rises to a certain value over 50°C, the oxidation reaction begins and aids further evaporation of moisture from the coal. The specific temperature range depends on the coal rank. The latent heat of moisture evaporation consumes the heat generated by self-heating of the coal pile. At $t = 4$ h, the net heating rate approaches zero, indicating that the heat produced by oxidation heat has been completely used for moisture evaporation at this time. Subsequently, internal heat generation in the coal becomes dominant and there is a rise in pile temperature.

## 4.3. Gas generation based on heat generation rate

In the experiments, the CO and $CO_2$ gas concentrations monitored in the chamber do not directly represent the absolute volumes of gas generated; however, they can be used as indicators because of the constant air flow into the chamber. Figure 8a,b shows the profiles of CO and $CO_2$ gas concentrations and temperatures at the centre of the WMB under supercritical and subcritical conditions, respectively. It is clear that these profiles exhibit almost the same trends with temperature and that CO and $CO_2$ gases are generated with the rise in temperature at the centre of the coal pile.

The oxidation reaction begins when the coal temperature rises to a value around 50°C that differs according to coal rank. Simultaneously, coal moisture suppresses the temperature rise owing to the latent heat of moisture evaporation taken from the coal particles. The moisture vapour moves to ambient air via weak natural convection flow through the pile. The CO concentration increases with the start of coal self-heating when the temperature rises beyond 60°C. Owing to the reduction in endothermic moisture evaporation with drying, the increased proportion of exothermic oxidative heat generation overcomes the endothermic drying.

In both cases, the coal pile is heated quickly by ambient air in the initial stages before $t = 4$ h, and no CO and $CO_2$ gas concentrations are detected. However, in the next stage during $t = 4$ to 16 h, the low concentrations of CO and $CO_2$ gases gradually increase with the slow temperature rise from 60°C, because the heat generated by oxidation is mainly consumed as latent heat of moisture evaporation.

In the subcritical case (figure 8a), the coal temperature $T_C$ reaches a peak ($T_C = 136°C$ at $t = 20.5$ h), while the CO and $CO_2$ concentrations also exhibit maxima. The oxidative reaction rate is also expected to reach a maximum at this peak temperature. When both the gas concentration and the central temperature exhibit peaks, the internal heat production and the heat dissipation to the ambient air eventually reach a dynamic equilibrium; subsequently, the heat dissipation gradually dominates in the subcritical case.

In the supercritical case, the CO and $CO_2$ concentrations increase sharply after $t = 15$ h with increasing central temperature. This state represents a self-heating reaction, leading to spontaneous combustion after the coal has completely dried out. Therefore, the intensity of oxidation after coal drying is sensitive whether the conditions are supercritical or subcritical.

## 4.4. Effects of coal particle size on the self-critical ignition process

Under the same conditions of ambient environment and pile volume, the coal particle size affects the self-heating process, because it affects the intensity of the internal natural convection flow, a key parameter governing the provision of oxygen and dissipation of heat in the pile. In this study, experiments were performed to clarify the influence of differences in particle sizes on $T_{CSIT}$.

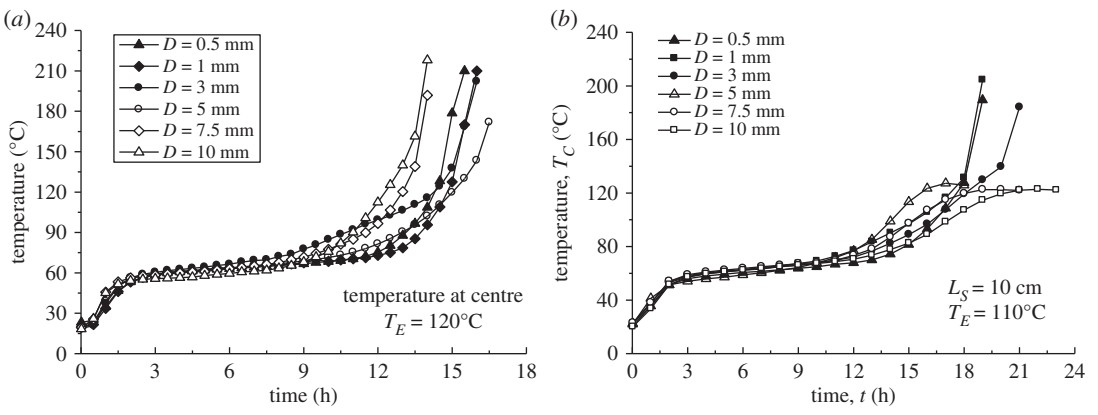

**Figure 9.** Temperature profiles for supercritical and subcritical $T_E$ with different particle sizes: (*a*) $L_S = 10$ cm, $T_E = 120°C$ (original wet coal); (*b*) $L_S = 10$ cm, $T_E = 110°C$ (original wet coal).

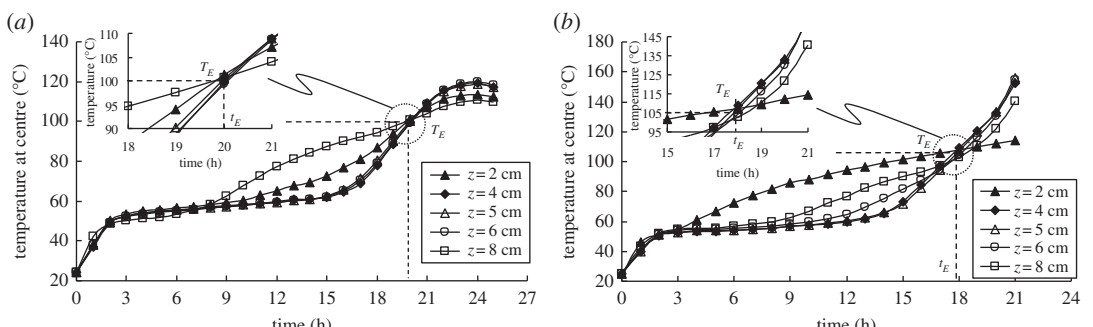

**Figure 10.** Comparisons of coal temperature profiles at different heights in the coal pile ($L_S = 10$ cm) in the subcritical and supercritical states: (*a*) subcritical state ($D = 1$ mm, $T_E = 100°C$, original wet coal); (*b*) supercritical state ($D = 1$ mm, $T_E = 105°C$, original wet coal).

Figure 9 shows the pile centre temperature–time curves for different coal particle sizes at $T_E = 120°C$, which yields the supercritical state. Coal piles for all particle sizes underwent spontaneous combustion; however, the time at which the self-ignition state was reached depended on particle size. The rate of temperature increase to 160°C was the fastest for coal particle size $D = 10$ mm, followed by 7.5, 0.5, 1, 3 and 5 mm. However, the temperature profile for $D = 3$ mm shows a contrasting behaviour to those for the other sizes, being higher in the initial stage before $t = 10$ h, but lower in the self-ignition stage. For smaller particle sizes, the apparent surface area is expected to result in an increase in the area that is in contact with air and from which coal moisture evaporates. The particles with $D = 10$ mm exhibit the fastest self-ignition. This is because the permeability of the coal pile is proportional to the square of the particle size (according to the Kozeny–Carman equation) [20]. The larger particle size results in a greater permeability of the pile, which induces a stronger natural convection flow, thereby contributing to faster drying of coal particles, allowing oxygen to enter the pile and leading to a supercritical state. Meanwhile, for smaller coal particles, natural convection is suppressed because of the reduced permeability. However, for a subcritical state, the effect of heat transfer from the coal pile to the ambient air suppresses the temperature rise in the pile. Therefore, the coal particle size has a complicated effect on both oxidation and heat dissipation processes.

## 4.5. Relation between the heating rate at the cross-point $T_C = T_E$ and supercriticality

Figure 10 compares coal temperature profiles before the cross-point ($T_C = T_E$) at different heights in the coal pile ($L_S = 10$ cm) in the subcritical and supercritical states. The elapsed time up to the cross-point (called the safety stock time), $t_E$, is almost the same for different heights in the coal pile. This means that temperatures in all areas of the coal pile are close to $T_E$ without inducing natural convection flow.

To consider the effect of the temperature gradient of the coal on supercriticality and subcriticality, we introduce a non-dimensional index $I_{HR}$ that is related to the representative coal pile length size

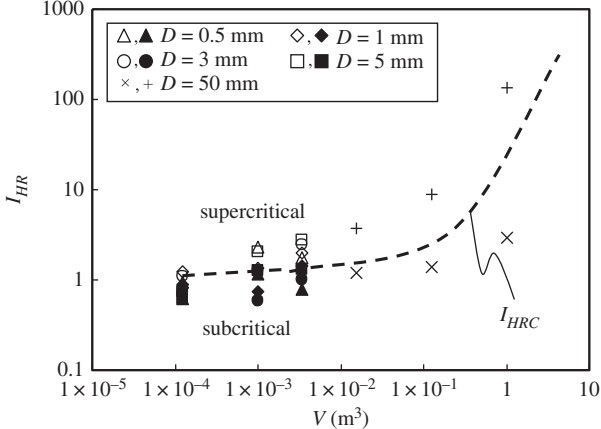

**Figure 11.** $I_{HR}$ boundary line between supercritical and subcritical states versus coal pile volume ($I_{HR} < I_{HRC}$: subcritical; $I_{HR} > I_{HRC}$: supercritical).

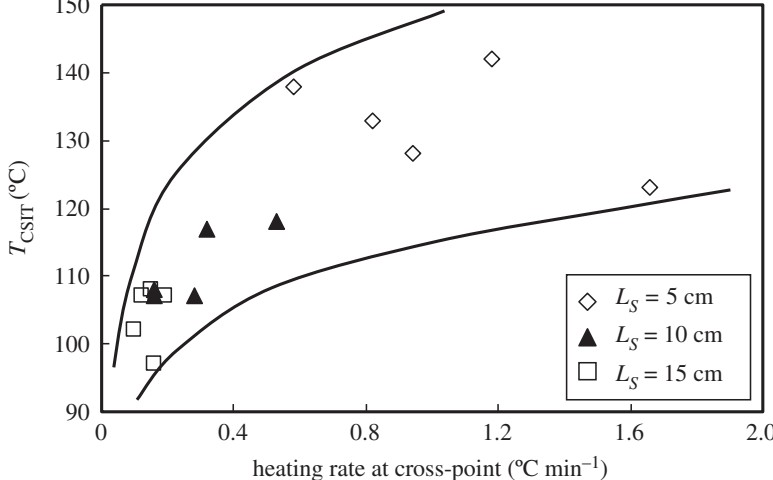

**Figure 12.** Relationship between the heating rate at the cross-point and the $T_{CSIT}$ of the coal pile.

$L$ (m), the ambient temperature $T_E$ (°C), the initial coal temperature $T_0$ (°C) and the thermal diffusivity of coal $a$ (m² s⁻¹):

$$I_{HR} = \frac{(\partial T_C / \partial t)_{T_C = T_E}}{\alpha \cdot (T_E - T_0)/L^2}. \tag{4.1}$$

The denominator in this equation normalizes the temperature gradient with respect to the temperature difference, the pile size and the thermal properties of the pile.

Based on the experimental results for the coal pile $T_{CSIT}$ and the heating environment temperature, the values of the index $I_{HR}$ are plotted versus coal pile volume $V$ in figure 11 to show whether the state of the coal pile is supercritical or subcritical. As the coal pile volume becomes larger, there is a more distinct separation between the supercritical and subcritical states, as defined by the line representing the boundary values of $I_{HR}$. It is found that these boundary values of $I_{HRC}$ are related exponentially to the coal pile volume $V$ (m³) as follows:

$$I_{HRC} = 1.48 \exp(3.1V) \tag{4.2}$$

## 4.6. Relation between heating rate at the cross-point and $T_{CSIT}$

The heating rate at the cross-point and the lead time up to the cross-point were measured in WMB tests to find the effect of particle size on the heating rate at the cross-point and the lead time of the coal pile. The corresponding relationships for different particle sizes are presented in figures 12 and 13. The heating rate at the cross-point of the coal pile is insensitive to the coal particle size. A greater heating rate at the cross-

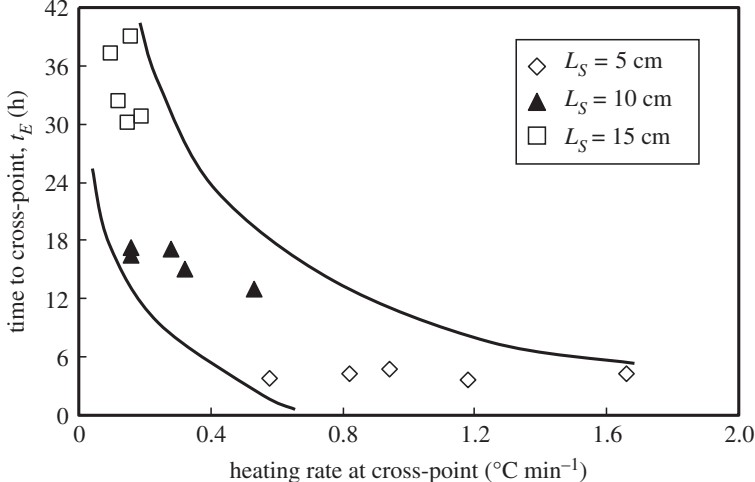

**Figure 13.** Relationship between the heating rate at the cross-point and the lead time to the cross-point.

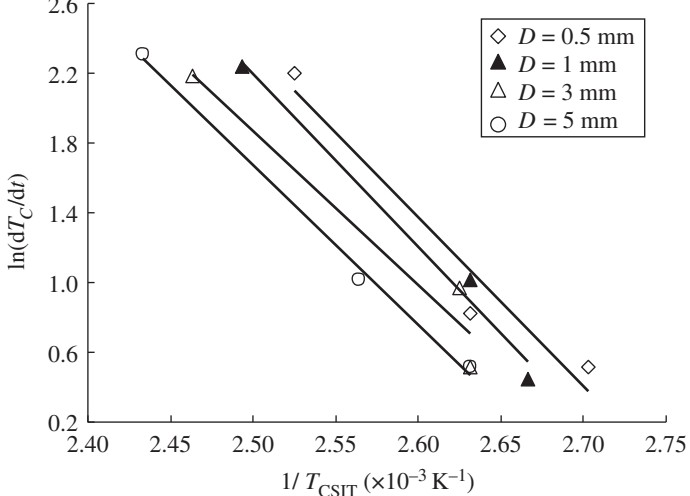

**Figure 14.** Plot of $dT_C/dt$ versus $T_K^{-1}$ ($E \approx 100$–$130$ J mol$^{-1}$).

point results in a higher heat generation rate in the coal pile; however, $T_{\text{CSIT}}$ also becomes higher. This means that a coal pile with a greater heating rate becomes more depleted after the cross-point and has less of a tendency to self-ignition.

As can be seen from figure 13, the lead time to the cross-point, $t_E$, is reduced significantly with decreasing pile volume, and also with the increasing heating rate at the cross-point. This clearly demonstrates that pile packing with larger coal particles results in a lower $T_{\text{CSIT}}$ but a longer lead time to reach $T_{\text{CSIT}}$.

Therefore, a greater heating rate at the cross-point with a shorter time is not a necessary condition for self-ignition of a coal pile, because it is found that the cooling rate after the cross-point is also larger when the pile temperature is reduced to the ambient temperature $T_E$.

## 4.7. Effects of coal particle size on apparent activation energy

The apparent activation energy $E$ is a useful parameter to evaluate the tendency to spontaneous combustion, because it has been reported that higher values of $T_{\text{CSIT}}$ have been measured for larger values of $E$ [22–24]. Figure 14 shows a semilogarithmic plot of $T_C/dt$ versus $T_K^{-1}$ for different particle sizes. The value of $E$ was calculated from the gradient of the line based on equation (2.1), and was estimated to lie in the following range:

$$E = 100\text{--}130 \text{ J mol}^{-1}. \tag{4.3}$$

According to the present results, the value of $E$ is not sensitive to particle size in the range 0.5–10 mm.

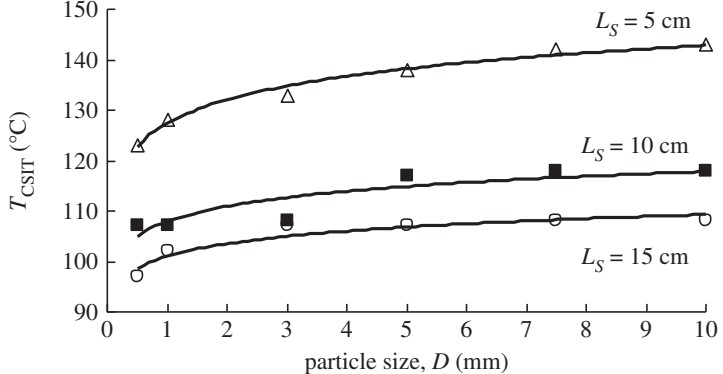

**Figure 15.** $T_{CSIT}$ versus coal particle size (original wet coal, $L_S = 5$, 10, 15 cm).

## 4.8. Effect of coal particle size on $T_{CSIT}$

Figure 15 shows the effect of coal particle size on $T_{CSIT}$. In a coal pile packed with larger particles, i.e. $D = 10$ and 7.5 mm, oxidative heat generation can be expected owing to the greater oxygen supply resulting from the higher velocity of the natural convection flow through piles with greater permeability. However, simultaneously, the more effective heat transfer from the coal particles to the flow transfers the heat of oxidation to the outer atmosphere, and $T_{CSIT}$ becomes higher than that of small particles.

A coal pile packed with small particles (0.5 and 1 mm) has smaller pores and weaker air convection resulting from the lower permeability, and there is greater retention of internal temperature in the pile, with its greater internal surface area and weaker convection air flow in the temperature range above $T_E$. The specific surface area and reduced porosity slow down the rise in temperature, resulting in a relatively low heating rate and a long lead time to reach the ambient temperature. The smaller the particle size, the lower is the permeability. This reduces oxygen penetration into the pile and the intensity of the oxidation reaction is reduced. However, heat transfer from the coal is also reduced and the heat of oxidation accumulates in the pile when this is greater than the heat loss. A coal pile with smaller particles has a larger internal surface area and the reaction surface between the coal and oxygen is large, so the reactions go further towards completion. Besides, the rate of loss of the heat generated by these reactions is relatively small, and the next reactions in the chain proceed continually even under the conditions of a limited oxygen supply. For large particles, although the amount of oxygen that penetrates into the pile is sufficient for reaction, the heat loss becomes greater. In this case, the heat generated does not accumulate. In addition, the larger the particles, the smaller is the effective reaction surface of the coal pile, which is not conducive to reaction of the coal.

The data plotted in figure 15 provide direct evidence that the larger the particles in the coal sample, the higher is the temperature for critical self-ignition. However, with increasing pile volume $V$ ($=Ls^3$), the influence of particle size on $T_{CSIT}$ is less than that of pile volume. For the $L_S = 5$ cm WMB, the difference between $T_{CSIT}$ for the 10 and 0.5 mm particles is approximately 20°C, while it reduces to 10°C for the $L_S = 10$ cm basket with an eight-times-larger volume. These results show that although the coal particle size affects $T_{CSIT}$, the basket volume has a more significant impact.

The empirical equation (1.1) estimating $T_{CSIT}$ in terms of coal pile volume $V$ ($=Ls^3$) (m³) has been revised by adding the effect of particle size as follows:

$$T_{CSIT} = 93 - 12.73 \cdot \log(V) + 10.9 \cdot \log\left(\frac{D}{L_S}\right). \tag{4.4}$$

This equation suggests that for the same coal components, the effect of pile volume $V$ is dominant, as suggested by Wang *et al.* [19], followed by the ratio of coal particle size to pile size, $D/L_S$. The ratio $D/L_S$ is used here because it gives a better match with the experimental results, with a correlation factor $R^2 = 0.91$, than when only $D$ is used in equation (4.4).

Figure 16 shows the relationship between the measured values of the coal pile $T_{CSIT}$ and the values calculated using equation (4.4). The data in the figure include previous measurements on the same coal sample excavated from Inner Mongolia using the larger WMBs, $L_S = 25$, 50 and 100 cm [19]. As expected,

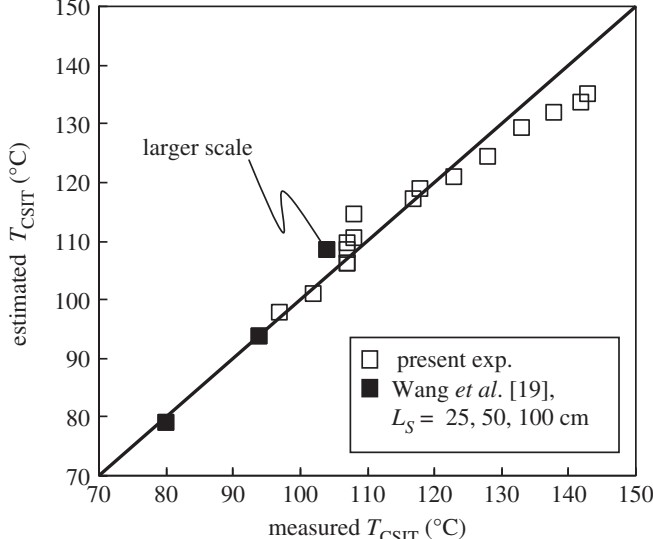

**Figure 16.** Relationship between measured values of $T_{CSIT}$ and those estimated using equation (4.4).

for coal piles consisting of different coal particle sizes equation (4.4), taking account of the effect of coal particle size, gives much better results than equation (1.1) when predicting the pile $T_{CSIT}$.

## 5. Conclusion

The effect of coal particle size on the critical self-ignition temperature $T_{CSIT}$ has been investigated for a coal pile packed with low-rank coal particles using the WMB test based on the F–K equation. The values of $T_{CSIT}$ and the apparent activation energy for different volumes of the piles packed with coal particles of different sizes from 0.5 to 10 mm have been measured.

The results of this study can be summarized as follows:

(1) For a constant WMB volume with $L_S = 10$ cm, $T_{CSIT}$ increases gradually with increasing coal particle size, because the natural convection flow facilitates the transport of heat from inside to outside owing to the greater permeability.
(2) Moisture in the coal prolongs the period before the start of self-heating. $T_{CSIT}$ measured by the WMB test for a coal pile with $L_s = 15$ cm packed with dried coal particles was reduced by approximately 10°C compared with the original wet coal.
(3) The apparent activation energy of the coal pile was measured as $E = 100$–$130$ J mol$^{-1}$ for $D = 0.5$–$10$ mm.
(4) Supercriticality or subcriticality of the coal was assessed using a non-dimensional index, $I_{HR}$, based on the temperature gradient at the temperature cross-point between coal and ambient temperatures for coal piles with various volumes and particle sizes. The critical value $I_{HRC}$ at the boundary between supercriticality and subcriticality was determined as a function of pile volume.
(5) The effect of the ratio of coal particle size to coal pile size, $D/L_S$, on $T_{CSIT}$ was incorporated into the original equation that gave $T_{CSIT}$ in terms of just coal pile volume. The modified equation may be useful to evaluate the effect of coal particle size on coal temperature rise in a coal pile.

Data accessibility. Experimental data are available within the Dryad repository, at https://doi.org/10.5061/dryad. gd4dg24 [25].
Authors' contributions. Y.W. conceived the original idea for this study and designed the study. X.Z. and K.S. prepared all samples for analysis, coordinated the study and helped draft the manuscript. Y.W. and H.Z. carried out the molecular laboratory work. Y.W., X.Z., H.Z. and K.S. collected and analysed the data. Y.W. wrote the manuscript. All authors gave final approval for publication.
Competing interests. We declare we have no competing interests.
Funding. Financial support came from the Japan Society for the Promotion of Science Grant-in-Aid for Scientific Research (grant no. 25303030).

Acknowledgements. This work was conducted as a joint project of the Liaoning Technical University and Kyushu University. The authors gratefully acknowledge the material support by the Institute of Engineering and Environment of Liaoning Technical University.

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
