## [Reviewer comments · Royal Society Open Science]

Review History

RSOS-190374.R0 (Original submission)

Review form: Reviewer 1

Is the manuscript scientifically sound in its present form?

Yes

Are the interpretations and conclusions justified by the results?

Yes

Is the language acceptable?

Yes

Is it clear how to access all supporting data?

Yes

Do you have any ethical concerns with this paper?

No

Have you any concerns about statistical analyses in this paper?

No

Recommendation?

Accept with minor revision (please list in comments)

Comments to the Author(s)

Journal: Royal Society Open Science

Manuscript ID: RSOS-190374

1. The number of references included in the abstract is relatively lack, and the literature on coal spontaneous combustion research should be increased for comparison.
2. Due to the font, some symbols in the manuscript, the superscript of the symbols, and on page 32 bottom are displayed incorrectly, which makes reading difficult. It is recommended to change the font to "times new roman".
3. How to obtain "Q" and "A" which are appeared in Eq. (3)?
4. Due to the heat transfer impediment caused by the solid state of the coal, is the calorimeter only measures whether the center of the coal is sufficient? If a large amount of coal accumulates, will it cause different exothermic conditions??
5. It can be seen from the Fig. 5, that coal has both endothermic and exothermic conditions. Does this phenomenon have an influence on subsequent evaluation?
6. What kind of reaction mode should coal spontaneous combustion belong to, nth order reaction or autocatalytic?
7. The article emphasizes the danger of spontaneous combustion of coal. However, there was not clearly pointed out how to reduce possibility hazard of spontaneous combustion of coal in Conclusion. It is suggested to add conclusions in the Conclusion part.

Review form: Reviewer 2

Is the manuscript scientifically sound in its present form?

No

Are the interpretations and conclusions justified by the results?

No

Is the language acceptable?

No

Is it clear how to access all supporting data?

Yes

Do you have any ethical concerns with this paper?

No

Have you any concerns about statistical analyses in this paper?

No

Recommendation?

Major revision is needed (please make suggestions in comments)

Comments to the Author(s)

The authors presented an experimental study on the temperature profile and self-ignition temperature of low-rank coal pile, and discussed the effect of coal particle size on the self-ignition temperature. The present experimental results may be helpful for understanding the spontaneous combustion of coal pile. My review comments are listed as follows:

1. The writing English and literature review in the introduction are poor. The expressions for Eqs. (5) and (6) are not normative. The authors should carefully revised the text and present up to date research progress in the fields of heat and mass transfer as well as spontaneous combustion of coal pile.
2. The dimensions on both sides of Eqs. (1) and (7) are not equal. The authors should revise their equations.
3. The explanation about the effect of coal particle size on the critical self-ignition temperature is not clear. The smaller particle size has smaller pores and lower permeability, while the authors stated that coal pile with small particle size has larger permeability. The authors also stated that porosity is not dominant, which is contradictory with their finding. It can be deduced that coal pile with small particle has lower permeability (low porosity) and weaker convective air flow, while it has larger surface area. The authors should explain the effect of these two aspects on the critical self-ignition temperature.

Decision letter (RSOS-190374.R0)

04-Jun-2019

Dear Dr WANG,

The editors assigned to your paper ("Effects of Temperature Gradient and Particle Size on Self Ignition Temperature of Low Rank Coal Excavated from Inner Mongolia, China") have now received comments from reviewers. We would like you to revise your paper in accordance with the referee and Associate Editor suggestions which can be found below (not including confidential reports to the Editor). Please note this decision does not guarantee eventual acceptance.

Please submit a copy of your revised paper before 27-Jun-2019. Please note that the revision deadline will expire at 00.00am on this date. If we do not hear from you within this time then it will be assumed that the paper has been withdrawn. In exceptional circumstances, extensions may be possible if agreed with the Editorial Office in advance. We do not allow multiple rounds of revision so we urge you to make every effort to fully address all of the comments at this stage. If deemed necessary by the Editors, your manuscript will be sent back to one or more of the original reviewers for assessment. If the original reviewers are not available, we may invite new reviewers.

When submitting your revised manuscript, you must respond to the comments made by the referees and upload a file "Response to Referees" in "Section 6 - File Upload". Please use this to

document how you have responded to the comments, and the adjustments you have made. In order to expedite the processing of the revised manuscript, please be as specific as possible in your response.

- Data accessibility

If you wish to submit your supporting data or code to Dryad (<http://datadryad.org/>), or modify your current submission to dryad, please use the following link:
<http://datadryad.org/submit?journalID=RSOS&manu=RSOS-190374>

- Competing interests

- Authors' contributions

- Acknowledgements

- Funding statement

Kind regards,

Alice Power

Editorial Coordinator

on behalf of R. Kerry Rowe (Subject Editor)

Comments to Author:

Reviewers' Comments to Author:

Reviewer: 1

Comments to the Author(s)

Journal: Royal Society Open Science

Manuscript ID: RSOS-190374

1. The number of references included in the abstract is relatively lack, and the literature on coal spontaneous combustion research should be increased for comparison.
2. Due to the font, some symbols in the manuscript, the superscript of the symbols, and on page 32 bottom are displayed incorrectly, which makes reading difficult. It is recommended to change the font to "times new roman".
3. How to obtain "Q" and "A" which are appeared in Eq. (3)?
4. Due to the heat transfer impediment caused by the solid state of the coal, is the calorimeter only measures whether the center of the coal is sufficient? If a large amount of coal accumulates, will it cause different exothermic conditions??
5. It can be seen from the Fig. 5, that coal has both endothermic and exothermic conditions. Does this phenomenon have an influence on subsequent evaluation?
6. What kind of reaction mode should coal spontaneous combustion belong to, nth order reaction or autocatalytic?
7. The article emphasizes the danger of spontaneous combustion of coal. However, there was not clearly pointed out how to reduce possibility hazard of spontaneous combustion of coal in Conclusion. It is suggested to add conclusions in the Conclusion part.

Reviewer: 2

Comments to the Author(s)

The authors presented an experimental study on the temperature profile and self-ignition temperature of low-rank coal pile, and discussed the effect of coal particle size on the self-ignition temperature. The present experimental results may be helpful for understanding the spontaneous combustion of coal pile. My review comments are listed as follows:

1. The writing English and literature review in the introduction are poor. The expressions for Eqs. (5) and (6) are not normative. The authors should carefully revised the text and present up to date

research progress in the fields of heat and mass transfer as well as spontaneous combustion of coal pile.

2. The dimensions on both sides of Eqs. (1) and (7) are not equal. The authors should revise their equations.

3. The explanation about the effect of coal particle size on the critical self-ignition temperature is not clear. The smaller particle size has smaller pores and lower permeability, while the authors stated that coal pile with small particle size has larger permeability. The authors also stated that porosity is not dominant, which is contradictory with their finding. It can be deduced that coal pile with small particle has lower permeability (low porosity) and weaker convective air flow, while it has larger surface area. The authors should explain the effect of these two aspects on the critical self-ignition temperature.

Editorial Office Comments to the Authors:

For more information about language-polishing services endorsed by the Royal Society, please follow the link below:

<https://royalsociety.org/journals/authors/language-polishing/>

Author's Response to Decision Letter for (RSOS-190374.R0)

See Appendices A - D.

RSOS-190374.R1 (Revision)

Review form: Reviewer 1

Is the manuscript scientifically sound in its present form?

Yes

Are the interpretations and conclusions justified by the results?

Yes

Is the language acceptable?

Yes

Do you have any ethical concerns with this paper?

No

Have you any concerns about statistical analyses in this paper?

No

Recommendation?

Accept as is

Comments to the Author(s)

Comments for RSOS-190374.R1

The manuscript has been revised suitably from the comments. I suggest that it can be published in Royal Society Open Science.

Review form: Reviewer 2

Is the manuscript scientifically sound in its present form?

Yes

Are the interpretations and conclusions justified by the results?

Yes

Is the language acceptable?

Yes

Do you have any ethical concerns with this paper?

No

Have you any concerns about statistical analyses in this paper?

No

Recommendation?

Accept as is

Comments to the Author(s)

The authors have revised their manuscript according to the review comments and presented detailed response to each comment. I have no more question on it, and it can be recommended for possible publication in Royal Society Open Science.

Decision letter (RSOS-190374.R1)

07-Aug-2019

Dear Dr Wang,

I am pleased to inform you that your manuscript entitled "Effects of Temperature Gradient and Particle Size on Self Ignition Temperature of Low Rank Coal Excavated from Inner Mongolia, China" is now accepted for publication in Royal Society Open Science.

Royal Society Open Science operates under a continuous publication model (<http://bit.ly/cpFAQ>). Your article will be published straight into the next open issue and this will be the final version of the paper. As such, it can be cited immediately by other researchers.

As the issue version of your paper will be the only version to be published I would advise you to check your proofs thoroughly as changes cannot be made once the paper is published.

Best regards,

on behalf of Professor R. Kerry Rowe (Subject Editor)
openscience@royalsociety.org

Associate Editor Comments to Author:

Your manuscript has now been seen and reviewed by both of the previous referees. Both reviewers are satisfied with the revisions made, and we are pleased to inform you that your manuscript has now been accepted for publication in Royal Society Open Science. Congratulations -- and thank you for taking the time to revise your manuscript.

Reviewer comments to Author:

Reviewer: 1
Comments to the Author(s)

The manuscript has been revised suitably from the comments. I suggest that it can be published in Royal Society Open Science.

Reviewer: 2
Comments to the Author(s)

The authors have revised their manuscript according to the review comments and presented detailed response to each comment. I have no more question on it, and it can be recommended for possible publication in Royal Society Open Science.

Appendix A

Point by point responses to reviewers

Dear editor,

We have substantially revised our manuscript based on the reviewers' valuable questions and comments. We also expanded part of experiments providing details in the current version. The point-by-point answers to reviewers' questions and comments are as follows:

Reviewer: 1

Comments to the Author(s)

Journal: Royal Society Open Science

Manuscript ID: RSOS-190374

1. The number of references included in the abstract is relatively lack, and the literature on coal spontaneous combustion research should be increased for comparison.
2. Due to the font, some symbols in the manuscript, the superscript of the symbols, and on page 32 bottom are displayed incorrectly, which makes reading difficult. It is recommended to change the font to "times new roman".
3. How to obtain "Q" and "A" which are appeared in Eq. (3)?
4. Due to the heat transfer impediment caused by the solid state of the coal, is the calorimeter only measures whether the center of the coal is sufficient? If a large amount of coal accumulates, will it cause different exothermic conditions??
5. It can be seen from the Fig. 5, that coal has both endothermic and exothermic conditions. Does this phenomenon have an influence on subsequent evaluation?
6. What kind of reaction mode should coal spontaneous combustion belong to, nth order reaction or autocatalytic?
7. The article emphasizes the danger of spontaneous combustion of coal. However, there was not clearly pointed out how to reduce possibility hazard of spontaneous combustion of coal in Conclusion. It is suggested to add conclusions in the Conclusion part.

Reviewer Comments Reply:

1. The number of references included in the abstract is relatively lack, and the literature on coal spontaneous combustion research should be increased for comparison.

Response: We are grateful for the positive comments regarding our paper. We have added several references on the comparison of coal spontaneous combustion research.

“Some of these approaches have focused on the study of the combustion products of the low-temperature oxidation process. From an examination of the mechanism of oxidation of coal of different particle sizes in the temperature range below 100 °C, Wang *et al.* [10] proposed a temperature threshold for thermal decomposition in low-temperature oxidation. Using techniques of organic chemistry and quantum chemistry, Xu *et al.* [11] studied heat generation by oxygen-containing functional groups during spontaneous combustion. They found that the presence of such groups is critical to heat production during the early stages of combustion and that this explained why the coal temperature rises slowly in this stage. Observations of raw and pyrolysed coal samples have shown that the active sites are influenced by the content of oxygen-containing functional groups, pyrolysis temperature, oxidation temperature, particle size and pyrolysis-oxidation times. Li *et al.* [12] proposed that two reaction processes are involved in coal oxidation: oxidation of active sites after coal pyrolysis and subsequent thermal decomposition with the generation of functional groups.

We believe that laboratory experiments are also important for analysis of the self-heating of coal, prediction of spontaneous combustion and design of fire extinguishing methods [13, 14].

2. Due to the font, some symbols in the manuscript, the superscript of the symbols, and on page 32 bottom are displayed incorrectly, which makes reading difficult. It is recommended to change the font to “times new roman”.

Response: Thank you for your precise suggestions. We revised the superscript of the symbols by changing to “Times new roman”.

3. How to obtain “Q” and “A” which are appeared in Eq. (3)?

Response: Thank you for your suggestion on Eq.(3).

We agree that we did not provide enough descriptions of Q and A in our initial manuscript. Our purpose to present this equation is to evaluate the activation energy, E , vs. particle size of the coal sample. Simultaneously, Q and A were calculated according to the Frank-Kamenetskii’s model. We provided the calculation procedure in the previous study (Wang et al., 2017)^[19].

Based on the Frank-Kamenetskii’s model, critical self-ignition temperature can be formulated based on heat balance between heat generation and heat loss rates of the coal pile at the temperature. Suppose the heat generation rate is expressed by the Arrhenius equation consisting of parameters for activation energy, E , frequency factor, A .

$$Q = \rho \cdot C_p \cdot \frac{dT_C}{dt} = A \cdot \exp\left[-\frac{E}{RT_K}\right] \quad (2)$$

$$\ln\left(\frac{dT_C}{dt}\right) = \ln\left(\frac{Q \cdot A}{\rho \cdot C_p}\right) - \frac{E}{R} \cdot \frac{1}{T_K} \quad (3)$$

By plotting $\ln(dT_C/dt)$ against $-T_K^{-1}$, $\ln(QA/\rho/C_p)$ and E/R are obtained from the values of the intercept and slope of the linear equation, respectively. The coal density ρ and specific heat capacity of coal C_p were measured before the experiment. The coal temperature T_K can be measured during the experiment. Finally, the heat generation rate Q and the frequency factor A can be obtained from equations (2) and (3).

We revised the manuscript by adding the sentence to explain it by using the reference [19].

4. Due to the heat transfer impediment caused by the solid state of the coal, is the calorimeter only measures whether the center of the coal is sufficient? If a large amount of coal accumulates, will it cause different exothermic conditions?

Response: Thank you for your encouraging comments and insightful suggestions.

The heat transfer impediment caused by the solid state of the coal particle will cause a certain delay of the temperature rise in the pile. We evaluated that the pile size/volume is main parameter and coal particle size is second parameter. We presented the Eq. (7) to evaluate the effects of two factors including large amount of coal accumulates. The time that center temperature rises to the ambient temperature is not so sensitive to the to the heat transfer impediment caused by the solid state, because the natural convection heat transfer has a larger influence on the coal pile temperature changes.

In order to avoid the influence of natural convection and other conditions, we chose the cubic wire mesh basket as the coal container. Since the square frame is completely symmetrical, the temperature gradient at the central position is mostly 0 compared with other region, therefore not so sensitive to the effective thermal conductivity and radiation heat transfer. This area can effectively reflect the temperature change process of coal piles and internal natural convection. Therefore, this study selects the central location as the main monitoring point. For the same conditions, we also considered the difference in the temperature rise process of coal piles under large volume conditions. In view of this, we also carried out experiments on 1m^3 coal samples (Wang et al., Determination of critical self-ignition temperature of low-rank coal using a 1 m wire-mesh basket and extrapolation to industrial coal piles. *Energ. Fuel* **2017**, 31, 6700-6710) to verify these differences. It is found that the internal exothermic conditions and temperature rise law are basically consistent with the small volume under large volume conditions.

As the reviewer's valuable comments, our final research objective will be to study the spontaneous combustion characteristics of the coal pile under larger volume conditions and analyze the variation law of the heating curve at different positions with combining numerical simulations to further improve the applicability of the theory.

5. It can be seen from the Fig. 5, that coal has both endothermic and exothermic conditions.

Does this phenomenon have an influence on subsequent evaluation?

Response: We are grateful for the kind comments regarding our paper.

We evaluated that the endothermic phenomenon, as shown in Fig. 5, does not have a big effect on supercritical or subcritical result of coal piles based on heat balance in the pile, because the endothermic heat is much less than exothermic one and generated in the lower temperature range. However, in the actual situation, we need to consider their effect on the leading time to reach the supercritical condition (start of spontaneous combustion), since the endothermic heat is mainly related to evaporation of coal moisture. Therefore, to control the coal pile heating process, the correct model to treat coal moisture behavior is needed in our next research.

Based on the above discussion, we add the sentence to the manuscript as follows:

“The endothermic phenomenon, as shown in Fig. 5, does not have a large effect on the supercriticality or subcriticality of a coal pile based on the heat balance in the pile, because the endothermic heat is much less than the exothermic heat and is generated in the lower- temperature range owing to evaporation of coal moisture.”

6. What kind of reaction mode should coal spontaneous combustion belong to, nth order reaction or autocatalytic?

Response: Thank you for your valuable comments.

At present, we are not able to answer appropriately to your question, because the spontaneous combustion process of coal includes complicated reactions from low temperature range and each temperature range shows different chemical and physical phenomena. The reaction process reaching to coal spontaneous combustion is characterized mainly two sequences of direct oxidation of coal and adsorption of coal oxygen have been confirmed. Previous researches, such as Qi (2011)[Qi, Oxidation and self- reaction of active groups in coal. *J. China Coal Soc.* **2011**, 36, 2133-2134.] used Fourier transform infrared spectroscopy to study the oxidation and self-reaction process of reactive

groups in coal, and clarified the distribution of groups in coal. And they also confirmed that some of the reactive groups can start reactions under anaerobic conditions. As the temperature of the coal rises continuously due to the physical changes in the initial period, the internal reaction process transitions from physical adsorption to chemical adsorption and generating chemisorptions heat. The chemical adsorption of the process will automatically accelerate into a chemical reaction, and produce reaction products such as CO₂, CO, H₂O, etc., with the oxidation reaction heat. The generation of heat in turn promotes further acceleration of the reaction, eventually resulting in spontaneous combustion.

Detailed interpretation of the effectiveness factor had been reported in many literatures [Hull et al., The role of the diffusion of oxygen in the ignition of a coal stockpile in confined storage. *Fuel*. **1997**, 76, 975-983]. The reaction order value in low temperature oxidation of coal and other carbonaceous materials has been indicated to vary from 0.5 to 1 [Carras and Young, Self-heating of coal and related materials: model application and test methods. *Prog. Energy Combust. Sci.* **1994**,20,1–15]. According to the published literatures, the vast majority of reaction orders are usually: $n = 1$, $n = 0.61$, $n = 0.52$, $n = 0.7$ and $n = 0.5$ [Taraba and Michalec, Effect of longwall face advance rate on spontaneous heating process in the gob area -CFD modeling. *Fuel*. **2011**, 90, 2790-2797; Yuan and Smith, CFD modeling of spontaneous heating in a large-scale coal chamber. *J. Loss. Prev. Process Ind.* **2009**, 22, 426–433; Fierro et al., Schmal, D. Model predictions and experimental results on self-heating prevention of stockpiled coals. *Fuel*. **2001**, 80, 125–134; Schmal et al., A model for the spontaneous heating of coal. *Fuel*. 1985, 64, 963–972; Zhang, J.W.; Choi et al., Modelling and parametric investigations on spontaneous heating in coal pile. *Fuel*. **2016**, 176, 181-189.]. In the case of the spontaneous combustion of coal, temperatures are extremely low and chemical kinetics play the main role on the reaction rate.

7. The article emphasizes the danger of spontaneous combustion of coal. However, there was not clearly pointed out how to reduce possibility hazard of spontaneous combustion of coal in Conclusion. It is suggested to add conclusions in the Conclusion part.

Response: Thanks for the useful comments. Your suggestions are very important and great significance to our research.

This study focused on the effects of ambient temperature and coal particle size on the critical self-ignition temperature of low-quality coal, and divides the critical state of coal samples. The coal status of supercritical or subcritical was related using with the non-dimensional index, I_{HR} , for temperature gradient at the temperature cross point between coal and environment temperatures for coal piles with various volumes and particle sizes. The current research is a basic research, which can predict and early warning the spontaneous combustion of coal piles under different conditions, such as coal pile volume and particle size. As the reviewers suggested, we would have provided how to reduce the risk of coal spontaneous combustion and prevent the spontaneous combustion of coal. However, to provide the actual measures to prevent hazards, understanding sensitivities of major parameters, such as pile volume and particle size is needed, because the spontaneous combustion of coal included many parameters. At present, we have begun to purchase relevant experimental equipment to show the sensitivity of parameters on the spontaneous combustion. Our next step is to conduct research on fire prevention technology, and we also plan to carry out field test. The prevention and control measures of coal spontaneous combustion will be further studied by combining lab experiments, field test and numerical simulation.

Reviewer: 2

Comments to the Author(s)

The authors presented an experimental study on the temperature profile and self-ignition temperature of low-rank coal pile, and discussed the effect of coal particle size on the self-ignition temperature. The present experimental results may be helpful for understanding the spontaneous combustion of coal pile. My review comments are listed as follows:

1. The writing English and literature review in the introduction are poor. The expressions for Eqs. (5) and (6) are not normative. The authors should carefully revised the text and present up to date research progress in the fields of heat and mass transfer as well as spontaneous combustion of coal pile.
2. The dimensions on both sides of Eqs. (1) and (7) are not equal. The authors should revise their equations.

3. The explanation about the effect of coal particle size on the critical self-ignition temperature is not clear. The smaller particle size has smaller pores and lower permeability, while the authors stated that coal pile with small particle size has larger permeability. The authors also stated that porosity is not dominant, which is contradictory with their finding. It can be deduced that coal pile with small particle has lower permeability (low porosity) and weaker convective air flow, while it has larger surface area. The authors should explain the effect of these two aspects on the critical self-ignition temperature.

Reviewer Comments Reply:

1. The writing English and literature review in the introduction are poor. The expressions for Eqs. (5) and (6) are not normative. The authors should carefully revised the text and present up to date research progress in the fields of heat and mass transfer as well as spontaneous combustion of coal pile.

Response: Thank you for your insightful suggestions.

As the reviewer suggestion, we have already polished the manuscript by a professional language editing company. Besides, we add the references and list the comparisons of coal spontaneous combustion in the introduction part as follows:

“1. Deng, J.; Zhao, J.Y.; Zhang, Y.N.; Huang, A.C.; Liu, X.R.; Zhai, X.W.; Wang, C.P. Thermal analysis of spontaneous combustion behavior of partially oxidized coal. *Process Saf. Environ.*, **2016**, 104, 218-224.”

Besides, we also add the references of research progress in the fields of coal pile heat and mass transfer as follows:

10. Wang, H.; Dlugogorski, B.Z.; Kennedy, E.M. Analysis of the mechanism of the low-temperature oxidation of coal. *Combust. Flame*. **2003**, 134, 107-117.
11. Xu, T.; Xie, Q.M.; Kang, Y.T. Heat effect of the oxygen-containing functional groups in coal during spontaneous combustion processes. *Adv. Powder Technol.* **2017**, 28, 1841-1848.
12. Li, J.H.; Li, Z.H.; Yang, Y.L.; Wang, C.J. Study on oxidation and gas release of active sites after low-temperature pyrolysis of coal. *Fuel*. **2018**, 233, 237-246.

13. Liang, Y.T.; Zhang, J.; Wang, L.C.; Luo, H.Z.; Ren, T. Forecasting spontaneous combustion of coal in underground coal mines by index gases: A review. *J. Loss Prevent. Proc.* 2019, 57, 208-222.
14. Syrodoy, S.V.; Kuznetsov, G.V.; Gutareva, N. Y.; Salomatov, V.V. The efficiency of heat transfer through the ash deposits on the heat exchange surfaces by burning coal and coal-water fuels. *J. Energy Inst.* **2017** (In press).

As you commented that Eqs. (5) and (6) are not normative, we could not present the normalized equations, since the spontaneous combustion in low temperature range includes complicated mass and heat transfer phenomenon and chemical reactions. Those equations are empirical equation based on experimental results to show the sensitivities of pile volume and particle size. We believe that those evaluations of the sensitivities are variable to understand of the spontaneous combustion of coal pile.

2. The dimensions on both sides of Eqs. (1) and (7) are not equal. The authors should revise their equations.

Response: Thank you for your encouraging comments and insightful suggestions. We revised the Eqs. (1) and (7) according to the Reviewer's comments

The Eqs.(1) and (7) are the empirical equations based on different scale experiments, because we can not clarify and the correct effects of other parameters. And the Eq.(1) only considers the critical self-ignition temperature, T_{CSIT} and pile volume, V . We believe that Eq.(7) includes a contribution or sensitivity of the coal particle size on T_{CSIT} with comparing the volume. As shown in Figure 15, the difference between T_{CSIT} for the 10 and 0.5 mm particles is approximately 20 °C in $L_S = 5$ cm WMB. But in $L_S = 10$ cm WMB it educed to 10 °C. This test result shows that the coal particle size affects its critical self-ignition temperature. However, it is clear that the basket volume has a larger effect on T_{CSIT} . The formulas presented the process to evaluate parameters' sensitivity for coal spontaneous combustions.

3. The explanation about the effect of coal particle size on the critical self-ignition temperature is not clear. The smaller particle size has smaller pores and lower permeability, while the authors stated that coal pile with small particle size has larger permeability. The authors also stated that porosity is not dominant, which is contradictory with their finding. It can be deduced that coal pile with small particle has lower permeability (low porosity) and weaker convective air flow, while it has larger surface area. The authors should explain the effect of these two aspects on the critical self-ignition temperature.

Response: Thank you for your comments.

Reviewer's comments are really correct. We agree that the smaller particle size has smaller pores and lower permeability. We are very sorry for our incorrect writing.

We revised the coal pile porosity statement in this section and added the explain effect of particle size and permeability on the critical self-ignition temperature as follows:

“Although the coal pile packing of small particle sizes (0.5 and 1 mm) has smaller pores and weaker air convections with **larger** permeability.....” to “coal pile packed with small particles (0.5 and 1 mm) has smaller pores and weaker air convection resulting from the lower permeability, and there is greater retention of internal temperature in the pile, with its greater internal surface area and weaker convection air flow in the temperature range above T_E .”

“The specific surface area and reduced porosity slow down the rise in temperature, resulting in a relatively low heating rate and a long lead time to reach the ambient temperature”

With the increase in the pile volume V , T_{CSIT} decreases for the same coal particle size. For the case of same coal pile volume, T_{CSIT} increases with the particle size. As the volume of the coal pile increases, the particle size range that affects the critical self-ignition temperature decreases. As show in Figure 15, the difference between T_{CSIT} for the 10 and 0.5 mm particles is approximately 20 °C for WMB with $L_S=5\text{cm}$, while it reduced to 10 °C in the $L_S = 10 \text{ cm}$ basket with eight-times-larger volume. For the case of WMB with $L_S = 10\text{cm}$ and 15 cm, T_{CSIT} tends to be stable when the coal particle size is larger than 5 mm and 3 mm respectably. These results indicated that the effect of particle size on the critical self-ignition temperature can be ignored when the coal pile is large enough.

To clear above discussion, we revised the manuscript as follows:

“.....The specific surface area and reduced porosity slow down the rise in temperature, resulting in a relatively low heating rate and a long lead time to reach the ambient temperature. The smaller the particle size, the lower is the permeability. This reduces oxygen penetration into the pile and the intensity of the oxidation reaction is reduced. However, heat transfer from the coal is also reduced and the heat of oxidation accumulates in the pile when this is greater than the heat loss. A coal pile with smaller particles has a larger internal surface area and the reaction surface between the coal and oxygen is large, so the reactions go further towards completion. Besides, the rate of loss of the heat generated by these reactions is relatively small, and the next reactions in the chain proceed continually even under the conditions of a limited oxygen supply. For large particles, although the amount of oxygen that penetrates into the pile is sufficient for reaction, the heat loss becomes greater. In this case, the heat generated does not accumulate. In addition, the larger the particles, the smaller is the effective reaction surface of the coal pile, which is not conducive to reaction of the coal.

The data plotted in figure 15 provide direct evidence that the larger the particles in the coal sample, the higher is the temperature.....”

Appendix B

List of revision parts

Our revision is general and not specified with data. The revisions of the paper are as follows,

1. Page 3, Line 3, add sentence “Whatever the mining method, coal is transported to the ground in large quantities, and because of the intrinsic properties of coal, this presents hazards in all processing stages after mining [1].”
2. Page 3, Line 14, add sentence “Some of these approaches have focused on the study of the combustion products of the low-temperature oxidation process. From an examination of the mechanism of oxidation of coal of different particle sizes in the temperature range below 100 °C, Wang *et al.* [10] proposed a temperature threshold for thermal decomposition in low-temperature oxidation. Using techniques of organic chemistry and quantum chemistry, Xu *et al.* [11] studied heat generation by oxygen-containing functional groups during spontaneous combustion. They found that the presence of such groups is critical to heat production during the early stages of combustion and that this explained why the coal temperature rises slowly in this stage. Observations of raw and pyrolysed coal samples have shown that the active sites are influenced by the content of oxygen-containing functional groups, pyrolysis temperature, oxidation temperature, particle size and pyrolysis-oxidation times. Li *et al.* [12] proposed that two reaction processes are involved in coal oxidation: oxidation of active sites after coal pyrolysis and subsequent thermal decomposition with the generation of functional groups.
We believe that laboratory experiments are also important for analysis of the self-heating of coal, prediction of spontaneous combustion and design of fire extinguishing methods [13, 14].”
3. Page5 line 6, revised the Equation (1).
4. Page6 line 19, add sentence “The coal density ρ and specific heat capacity of coal C_p were measured before the experiment. The coal temperature T_K can be measured during the experiment. Finally, the heat generation rate Q and the frequency factor A can be obtained from

- equations (2) and (3) [19].”
5. Page11 line 14, add sentence “The endothermic phenomenon, as shown in Fig. 5, does not have a large effect on the supercriticality or subcriticality of a coal pile based on the heat balance in the pile, because the endothermic heat is much less than the exothermic heat and is generated in the lower- temperature range owing to evaporation of coal moisture.”
 6. Page23 line 5, revised the Equation (5).
 7. Page23 line 7, change “**Figure 11.** I_{HR} boundary line between super critical and subcritical vs. coal pile volume” to “**Figure 11.** I_{HR} boundary line between supercritical and subcritical states versus coal pile volume ($I_{HR} < I_{HRC}$: subcritical; $I_{HR} > I_{HRC}$: supercritical).”
 8. Page25 line 12, revised the Equation (6).
 9. Page 26 line 8, change sentence “Although the coal pile packing of small particle sizes (0.5 and 1 mm) has smaller pores and weaker air convections with **larger** permeability.....” to “A coal pile packed with small particles (0.5 and 1 mm) has smaller pores and weaker air convection resulting from the lower permeability, and there is greater retention of internal temperature in the pile, with its greater internal surface area and weaker convection air flow in the temperature range above T_E .”
 10. Page 26 line 11, change sentence “Despite it having a relatively small particle size, its specific surface area and porosity are not dominant.....” to “The specific surface area and reduced porosity slow down the rise in temperature, resulting in a relatively low heating rate and a long lead time to reach the ambient temperature.”
 11. Page 26 line 13, add sentence “The smaller the particle size, the lower is the permeability. This reduces oxygen penetration into the pile and the intensity of the oxidation reaction is reduced. However, heat transfer from the coal is also reduced and the heat of oxidation accumulates in the pile when this is greater than the heat loss. A coal pile with smaller particles has a larger internal surface area and the reaction surface between the coal and oxygen is large, so the reactions go further towards completion. Besides, the rate of loss of the heat generated by these reactions is relatively small, and the next reactions in the chain proceed continually even under the conditions of a limited oxygen supply. For large particles, although the amount of oxygen

that penetrates into the pile is sufficient for reaction, the heat loss becomes greater. In this case, the heat generated does not accumulate. In addition, the larger the particles, the smaller is the effective reaction surface of the coal pile, which is not conducive to reaction of the coal.”

12. Page27 line 10, revised the Equation (7).

13. Page30 line10, add reference

“1. Deng, J.; Zhao, J.Y.; Zhang, Y.N.; Huang, A.C.; Liu, X.R.; Zhai, X.W.; Wang, C.P. 2016 Thermal analysis of spontaneous combustion behavior of partially oxidized coal. *Process Saf. Environ.* **104**, 218-224.”

“2. Bhoi, S.; Banerjee, T.; Mohanty, K. 2014 Molecular dynamic simulation of spontaneous combustion and pyrolysis of brown coal using ReaxFF. *Fuel*. **136**, 326-333.”

“3. Qi, G.S.; Wang, D.M.; Chen Y.; Xin, H.H.; Qi, X.R.; Zhong, X.X. 2014 The application of kinetics based simulation method in thermal risk prediction of coal. *J. Loss Prevent. Proc.* **29**, 22-29.”

“4. Chen, B.; Diao, Z.J.; Lu, H.Y. 2014 Using the ReaxFF reactive force field for molecular dynamics simulations of the spontaneous combustion of lignite with the Hatcher lignite model. *Fuel*, **116**, 7-13.”

“5. Zhang, D.M.; Chu, Y.P.; Li, S.J.; Yang, Y.S.; Ye, C.; Wen, D.C. 2018 Petrophysical characterization of high-rank coal by nuclear magnetic resonance: a case study of the Baijiao coal. *Roy. Soc. Open Sci.* **339**, 102-110.”

“8. G, J.; Wen, H.; Zheng, X.Z.; Liu, Y.; Cheng, X.J. 2019 A method for evaluating the spontaneous combustion of coal by monitoring various gases. *Process Saf. Environ.* **126**, 223-231.”

“9. Li, Q.W.; Xiao,Y.; Wang, C.P.; Deng, J.; Shu, C.M. 2019 Thermo kinetic characteristics of coal spontaneous combustion based on thermogravimetric analysis. *Fuel* **250**, 235-244.”

“10. Wang, H.; Dlugogorski, B.Z.; Kennedy, E.M. 2013 Analysis of the mechanism of the low-temperature oxidation of coal. *Combust. Flame.* **134**, 107-117.”

“11. Xu, T.; Xie, Q.M.; Kang, Y.T. 2017 Heat effect of the oxygen-containing functional groups

in coal during spontaneous combustion processes. *Adv. Powder Technol.* **28**, 1841-1848.”

“12. Li, J.H.; Li, Z.H.; Yang, Y.L.; Wang, C.J. 2018 Study on oxidation and gas release of active sites after low-temperature pyrolysis of coal. *Fuel.* **233**, 237-246.”

“13. Liang, Y.T.; Zhang, J.; Wang, L.C.; Luo, H.Z.; Ren, T. 2019 Forecasting spontaneous combustion of coal in underground coal mines by index gases: A review. *J. Loss Prevent. Proc.* **57**, 208-222.”

“14. Syrodoy, S.V.; Kuznetsov, G.V.; Gutareva, N. Y.; Salomatov, V.V. 2017 The efficiency of heat transfer through the ash deposits on the heat exchange surfaces by burning coal and coal-water fuels. *J. Energy Inst.* (In press).”

“25. Wang, Y.J.; Zhang, X.M.; Zhang, H.M.; Sasaki, K. 2019 Data from: Effects of temperature gradient and particle size on self-ignition temperature of low-rank coal excavated from Inner Mongolia, China. Dryad Digital Repository. (doi:10.5061/dryad.gd4dg24)”

EDITORIAL CERTIFICATE

This document certifies that the manuscript below was edited for correct English language usage, grammar, punctuation and spelling by qualified native English speaking editors at The Charlesworth Group.

Paper Title:

Effects of Temperature Gradient and Particle Size on Self-ignition Temperature of Low-rank Coal Excavated from Inner Mongolia, China

Author:

Yongjun WANG

Date certificate issued:

July 05, 2019

cwauthors.com

Appendix D

FIGURES:

Figure 1. Cubic coal pile

(a)

Figure 2. Estimation of T_{CSIT} based on the pile temperature profile at constant ambient temperature: (a) temperature profile; (b) heating rate.

Figure 3. DTA system and constant temperature chamber

Figure 4. Experimental apparatus and temperature and gas monitoring system for the WMB test.

Figure 5. DTA curve of the coal sample used in this study (Original wet)

Figure 6. Examples of temperature profile at pile centre versus time ($L_S = 5$ cm, initial coal temperature = 25°C): (a) stepwise increase in ambient temperature (20°C every 2 h); (b) instantaneous setting of a constant ambient temperature from room temperature.

(a)

(b)

Figure 7. Comparison of heating-rate–time profiles between (a) original wet and (b) dried coal samples.

(a)

(b)

Figure 8. Gas concentration and heat generation rate for different ambient temperatures ($L_S = 10$ cm, $D = 5$ mm, original coal): (a) $T_E = 115^\circ\text{C}$ (subcritical); (b) $T_E = 120^\circ\text{C}$ (supercritical).

(a)

(b)

Figure 9. Temperature profiles for supercritical and subcritical T_E with different particle sizes: (a) $L_S = 10$ cm, $T_E = 120^\circ\text{C}$ (original wet coal); (b) $L_S = 10$ cm, $T_E = 110^\circ\text{C}$ (original wet coal).

(a)

(b)

Figure 10. Comparisons of coal temperature profiles at different heights in the coal pile ($L_S = 10$ cm) in the subcritical and supercritical states: (a) subcritical state ($D = 1$ mm, $T_E = 100^\circ\text{C}$, original wet coal); (b) supercritical state ($D = 1$ mm, $T_E = 105^\circ\text{C}$, original wet coal).

Figure 11. I_{HR} boundary line between supercritical and subcritical states versus coal pile volume ($I_{HR} < I_{HRC}$: subcritical; $I_{HR} > I_{HRC}$: supercritical).

Figure 12. Relationship between the heating rate at the cross-point and the T_{CSIT} of the coal pile.

Figure 13. Relationship between the heating rate at the cross-point and the lead time to the cross-point.

Figure 14. Plot of dT_c/dt versus T_K^{-1} ($E \approx 100-130$ J/mol).

Figure 15. T_{CSIT} versus coal particle size (original wet coal, $L_S = 5, 10, 15$ cm).

Figure 16. Relationship between measured values of T_{CSIT} and those estimated using equation (7).

TABLES:

Table 1. Properties of coal sample.

Analysis	Content	Original wet	Dry
Proximate analysis	Ash (%)	8.3	33.42
	Fixed carbon (%)	24.7	34.92
	Volatile Matter (%)	39.8	31.66
	Moisture (%)	27.2	0
Ultimate analysis	C (%)	70.44	32.42
	H (%)	4.91	2.26
	O (%)	21.39	9.76
	N (%)	1.35	0.62
	S (%)	1.91	0.53
Physical properties	Density, ρ (kg/m ³)	1300	1560
	Heat capacity, C_p (J/kg/°C)	2720	1740
	Heat conductivity, λ (Wm/°C)	0.21	0.07
	Thermal diffusivity, α (m ² /s)	5.94×10^{-8}	2.58×10^{-8}